# First Dye-Decolorizing Peroxidase from an Ascomycetous Fungus Secreted by *Xylaria grammica*

**DOI:** 10.3390/biom11091391

**Published:** 2021-09-21

**Authors:** Virginia Kimani, René Ullrich, Enrico Büttner, Robert Herzog, Harald Kellner, Nico Jehmlich, Martin Hofrichter, Christiane Liers

**Affiliations:** 1Unit of Environmental Biotechnology, International Institute Zittau, Dresden University of Technology, Markt 23, 02763 Zittau, Germany; virwkim@gmail.com (V.K.); rene.ullrich@tu-dresden.de (R.U.); enrico.buettner@tu-dresden.de (E.B.); robert.herzog@tu-dresden.de (R.H.); harald.kellner@tu-dresden.de (H.K.); martin.hofrichter@tu-dresden.de (M.H.); 2Kenya Industrial Research and Development Institute, Nairobi P.O. Box 30650-00100, Kenya; 3Helmholtz-Centre for Environmental Research–UFZ, Department of Molecular System Biology, 04318 Leipzig, Germany; nico.jehmlich@ufz.de

**Keywords:** dye-decolorizing peroxidase, *Xylaria grammica*, ascomycete, Mn^2+^ oxidation, Mn^2+^ binding site

## Abstract

Background: Fungal DyP-type peroxidases have so far been described exclusively for basidiomycetes. Moreover, peroxidases from ascomycetes that oxidize Mn^2+^ ions are yet not known. Methods: We describe here the physicochemical, biocatalytic, and molecular characterization of a DyP-type peroxidase (DyP, EC 1.11.1.19) from an ascomycetous fungus. Results: The enzyme oxidizes classic peroxidase substrates such as 2,6-DMP but also veratryl alcohol and notably Mn^2+^ to Mn^3+^ ions, suggesting a physiological function of this DyP in lignin modification. The K_M_ value (49 µM) indicates that Mn^2+^ ions bind with high affinity to the *Xgr*DyP protein but their subsequent oxidation into reactive Mn^3+^ proceeds with moderate efficiency compared to MnPs and VPs. Mn^2+^ oxidation was most effective at an acidic pH (between 4.0 and 5.0) and a hypothetical surface exposed an Mn^2+^ binding site comprising three acidic amino acids (two aspartates and one glutamate) could be localized within the hypothetical *Xgr*DyP structure. The oxidation of Mn^2+^ ions is seemingly supported by four aromatic amino acids that mediate an electron transfer from the surface to the heme center. Conclusions: Our findings shed new light on the possible involvement of DyP-type peroxidases in lignocellulose degradation, especially by fungi that lack prototypical ligninolytic class II peroxidases.

## 1. Introduction

Peroxidases of the DyP-type or more shortly DyPs (dye-decolorizing peroxidases, EC 1.11.1.19) were first reported in 1995 for cultures of *Bjerkandera adusta* (originally described as a strain of *Geotrichum candidum* [1,2]). Later, DyPs were recognized as a new family of heme peroxidases found in both fungi and bacteria. Thus, there are features in the secondary and tertiary structure of DyPs that do not allow them to be classified into any of the known peroxidase groups of bacteria, fungi, or plants (class I, II, and III peroxidases/PODs, respectively). Sequence similarities to ligninolytic class II peroxidases are low (0.5–5%) and the typical heme-binding region, which is conserved among the whole catalase-peroxidase superfamily, does not contain the distal His [3,4,5]. All DyPs contain a highly conserved GXXDG motif, and the distal His is replaced by an aspartate (Asp) as an acid–base catalyst, which is assisted in proton acceptation and charge stabilization by an Arg [6].

Since their discovery, nine fungal DyPs (from among others *Mycetinis scorodonius* (*Msc*DyP), [7]; *Auricularia auricula-judae* (*Aau*DyP1, *Aau*DyP2), [8]; *Exidia glandulosa* (*Egl*DyP), *Mycena epipterygia* (*Mep*DyP), [9]) and fifteen bacterial enzymes (e.g., from *Thermobifidia fusca*, [10]; *Rhodococcus jostii*, [11]) have been described. Among them are eight wild-type proteins and three recombinant enzymes (one from *A. auricula judae* (r*Aau*DyP) and two from *Pleurotus ostreatus* (r*Pos*DyP), [12]). Wild-type proteins (e.g., *Aau*DyP1 and 2) can be produced in complex plant-based media such as diluted tomato juice or soybean meal suspension with or without the addition of ‘elicitors’ (e.g., guaiacol or β-carotene). Activity levels range from 100 to 8000 U L^−1^ (corresponding to 0.25 to 20 mg L^−1^ DyP protein, [8,9]). The production of recombinant fungal DyPs, e.g., of r*Bad*DyP in *Aspergillus oryzae* or of r*Aau*DyP and r*Pos*DyP in *Escherichia coli*, has facilitated the performance of mutational studies, increasing our understanding of the relationship between DyP structure and function [12,13,14,15]. So far, only three fungal DyPs (from *B. adusta*, *A. auricula-judae* and *P. ostreatus*) have been thoroughly characterized from the structural and mechanistic point of view [6,12,13,14,15,16,17,18].

DyPs oxidize a range of substrates, notably recalcitrant azo and anthraquinone dyes (e.g., Reactive Blue5, [2,9]), phenols (e.g., methoxy- and nitrophenols, [9,17]), terpenoids (e.g., β-carotene, [7]), non-phenolic aromatics (e.g., trimethoxybenzene, veratryl alcohol, [9]) and even the lignin model dimer ‘adlerol’ [8,9,15]. The oxidation of non-phenolic aromatics and adlerol works best at rather low pH (i.e., pH < 3.0, [5,8,9]). The specific activity of DyP towards adlerol at pH 3.0 was found to be one order of magnitude lower than that of LiP (lignin peroxidase) of *Phanerochaete chrysosporium*, but demonstrates that in theory, it may have the capacity to oxidize lignin or lignin-like molecules [9].

Furthermore, a few bacterial DyPs (e.g., *Rjo*DyP, [11]) and one recombinant fungal DyP (r*Pos*DyP4) were found to oxidize Mn^2+^ into Mn^3+^, an exclusive catalytic feature of manganese (MnPs) and versatile peroxidases (VP) [5]. Indeed, for one recombinant bacterial DyP from *R. jostii*, it has been shown that Mn^2+^ oxidation is involved in the oxidation and partial breakdown of adlerol (*k*_cat_ = 7.4 × 10^−3^; [11])—a convincing indication for the possible involvement of DyPs in lignin oxidation.

The physicochemical properties of fungal DyPs resemble those of class II PODs. They are glycosylated proteins (up to 20% sugars) that show a Soret band at 405–407 nm, reflecting the proximal heme-imidazole (His) ligation. Molecular weights of DyPs range between 40 and 67 kDa with isoelectric points (pI) between 3.5 and 4.3 [5]. According to the InterPro database (http://www.ebi.ac.uk/interpro/ accessed on 18 September 2021), the DyP family currently comprises over 5019 hypothetical proteins, of which 4886 are from bacteria, 122 from eukaryotes, and 11 from archaea [19], confirming their ubiquitous distribution but also the dominance of bacterial DyPs. Consequently, van Bloois and colleagues [20] suggested renaming the DyP family as the ‘bacterial heme peroxidase’ family. However, this suggestion was not accepted by the scientific community [21].

According to similarities in the primary structure (i.e., sequence homology), DyPs can be phylogenetically categorized in four classes/subfamilies [22]. Classes A, B, and C correspond to DyPs of prokaryotic origin (e.g., class A—*T. fusca* DyP, class B—*Schewanella oneidenis* DyP, class C—*Anabaena* and *Amycolatopsis* DyP); fungal DyPs (those belonging to the basidio- and ascomycetes) cluster only within class D and show between 7 to 16% homology to the three bacterial classes [21]. To overcome the ambiguity of DyP categorization, especially regarding classes C and D, a new classification has been proposed by Yoshida and Sugano [21]. It distinguishes between primitive (P), intermediate (I), and advanced (V) clades depending on both primary and particularly tertiary structure homologies. The V clade now includes the former classes C and D since both protein subfamilies appear to be more closely related according to structural homologies.

Although the DyP-catalyzed oxidation of lignin model compounds (adlerol, methoxybenzenes, [8,9]) and the enhancement of enzymatic straw hydrolysis by *Irpex lacteus* DyP [23] have been proven, the natural function of these enzymes remains unclear [15]. Interestingly, Sugawara et al. [24] have recently reported that a DyP of *Bjerkandera adusta* is produced and secreted in response to alizarin, an anti-fungal anthraquinone compound produced by the plant *Rubia tinctorum* (common madder). This indicates that DyPs could be part of the biochemical defense of fungi against toxic plant and microbial metabolites. The widespread occurrence of DyPs in forest soils (which has been shown at the transcript and activity levels) may support this assumption [25].

In the present study, we describe for the first time an Mn^2+^-oxidizing DyP-type peroxidase secreted by an ascomycetous fungus, a new strain of *Xylaria grammica* (IHIA82, GenBank accession number MK408621) collected from rotting plant debris in the Kakamega Forest National Reserve (Kenya; lat 0.33431, long 34.87814). The genome of this strain was sequenced with a total size of 47.0 Mbp and 12,126 predicted genes [26].

## 2. Materials and Methods

### 2.1. Screening and Production of a DyP-Type Peroxidase by X. grammica

For the production of the DyP-type peroxidases from *X. grammica*, liquid cultivation was performed using a glucose–peptone medium (SPM; glucose 28 g L^−1^, malt extract 3 g L^−1^, soy-peptone 12 g L^−1^, yeast extract 3 g L^−1^), which is known to stimulate the secretion of certain fungal enzymes [27,28]. A complete fungal pre-culture grown on malt-extract agar (MA) was homogenized in a 100 mL Erlenmeyer flask containing 80 mL sterile water using an Ultra-Turax^TM^ device; 9 mL of this homogenized suspension was transferred with a sterile pipette to 500 mL culture flasks containing 200 mL of the SPM. After inoculation, the cultures were agitated on a rotary shaker at 100 rpm and 23 °C. Enzyme activities (Mn^2+^-independent peroxidase activities) and pH were determined every 2–3 days over a total cultivation period of three weeks. Later, the production of DyP from *X. grammica* was performed on a larger scale in 500 and 1000 mL flasks containing 200 or 1000 mL of the SPM medium, respectively.

### 2.2. Enzyme Assays

During fungal cultivation, activities of an Mn^2+^-independent peroxidase (MiP) were measured to determine the most appropriate time for harvesting the cultures. All activity measurements were performed, after removing the fungal mycelium by centrifugation, in 1 mL cuvettes at room temperature using a Cary 50 UV/Vis spectrophotometer (Varian, Darmstadt, Deutschland), and the enzymatic activities were calculated in units. One unit is defined as that amount of enzyme that catalyzes the conversion or formation of 1 µmol substrate or product, respectively, per minute (µmol min^−1^).

To determine MiP activity, ABTS (0.3 mM final concentration) oxidation was followed in 50 mM sodium citrate buffer (pH 4.5) in the absence (for laccase) or presence of H_2_O_2_ (0.1 mM for MiP). Formation of the product was monitored at 420 nm (ε_420_ = 36 mM^−1^ cm^−1^, [8,29]). Phenol-oxidizing activity using 2,6-dimethoxyphenol (2,6-DMP; ε_469_ = 27.5 mM^−1^ cm^−1^) and the oxidation of synthetic dyes with Reactive Blue 5 (RBlue5, ε_598_ = 8.0 mM^−1^ cm^−1^) was measured in sodium citrate buffer (50 mM, pH 4.5) [9]. The oxidation of Mn^2+^ to Mn^3+^ was used to determine the manganese-dependent activity of the *X. grammica* peroxidase (manganese-oxidizing activities in fungal cultures are mainly due to MnP (EC 1.11.1.13) but may be also caused by VP (EC 1.11.1.16)) [30]. The formation of Mn^3+^–malonate complexes was monitored in the presence of MnCl_2_ (0.5 mM final concentration) in 50 mM sodium malonate buffer (pH 4.5) by measuring the initial increase in absorbance at 270 nm (ε_270_ = 11.5 mM^−1^ cm^−1^) after addition of H_2_O_2_ (0.1 mM final concentration). Veratryl alcohol (VA, ε_270_ = 9.3 mM^−1^ cm^−1^) oxidizing activity was measured in sodium tartrate buffer (50 mM, pH 3.0) in the presence of H_2_O_2_ (0.1 mM final concentration) [9].

### 2.3. Purification of the DyP from X. grammica (XgrDyP)

After reaching maximal activities, the culture liquids were harvested. The fungal mycelium was removed by filtration and the crude liquid was frozen at −20 °C prior to purification to remove polysaccharides by precipitation. After thawing, the crude enzyme liquid was again filtered at 11 °C (GF6, Sartorious GmbH, Göttingen, Deutschland) followed by ultrafiltration and dialysis steps using a tangential flow cassette (Vivaflow 200, cut-off 10 kDa; Sartorius, Göttingen, Germany). All chromatographic steps were performed with ÄKTA™ Avant FPLC systems (GE Healthcare, Chicago, IL, USA). Absorbing material eluting from the columns was simultaneously monitored at 280 nm (total protein) and 405 nm (heme proteins including DyP).

The crude peroxidase preparation of *X. grammica* (*Xgr*DyP) was purified by four steps of FPLC using anion exchange chromatography (AEX) and size exclusion chromatography (SEC) for separation. In the first and second steps, the concentrated crude enzyme was applied to a Q-Sepharose (26 × 100 mm) column and eluted with a linear gradient of 0–1.0 M NaCl in 10 mM sodium acetate buffer (pH 6.0) at a flow rate of 13 mL min^−1^ and with a fraction size of 7.0 mL. Peroxidase-positive fractions were loaded onto an SEC column HiLoad 26/600 Superdex 75 PG equilibrated with 50 mM sodium acetate buffer (pH 6.75) containing 0.1 M NaCl. Enzyme protein was eluted with the same buffer at a flow rate of 2.5 mL/min^−1^. The third step was performed on a MonoQ column (5 × 50 mm) with a NaCl gradient of 0–0.4 M in 10 mM sodium acetate (pH 6.0) and with a fraction size of 1.5 mL at a flow rate of 6 mL min^−1^. The last step was a re-chromatograhic elution of the peroxidase-positive fractions originating from the previous AEX separations. The parameters were the same as already described, except that the loading and elution buffers were adjusted to pH 5.8 and 6.0, respectively. Fractions containing *Xgr*DyP activity were pooled, concentrated, and washed with 10 mM sodium acetate buffer (pH 6.5).

### 2.4. Protein Determination

Total protein of crude extracts and of each purification step were determined using the method of Bradford [31] with a Roti^®^ Nanoquant Kit (Roth, Karlsruhe, Germany) and bovine serum albumin (BSA) as protein standards. Samples were pipetted into a 96-well microtiter plate in triplicate and the absorbance was measured at 590/450 nm with a microplate reader (Infinite 200, Tecan, Switzerland).

### 2.5. Protein Electrophoresis

SDS-PAGE was performed according to the protocol described in [32] with a vertical electrophoresis system (XCell *SureLock*^TM^ Mini-Cell, Invitrogen, Carlsbad, CA, USA) and Novex^®^NuPAGE^®^ 12% Bis-Tris Gels (Invitrogen, Waltham, MA, USA). This electrophoresis method was applied to check the protein purity under denaturing conditions and to determine the molecular weight of the purified enzymes. A low-MW protein mixture was used as standard (MBI Fermentas, St. Leon-Rot, Germany). Electrophoretically separated proteins were visualized by using a Colloidal Blue Staining Kit (Invitrogen, Waltham, MA, USA).

The above-mentioned electrophoresis system was also used to separate proteins according to their isoelectric points (p*I*s). Electrophoresis was performed according to the manufacturer’s protocol with Novex^®^ Vertical IEF Gels (pH gradient 3–7; Invitrogen, Waltham, MA, USA) and a Liquid Mix IEF Marker (pH 3.0–10.0; Serva, Heidelberg, Germany) as a reference. Protein bands in the vertical polyacrylamide gel were afterwards visualized by using a Colloidal Blue Staining Kit (Invitrogen, Waltham, MA, USA).

### 2.6. Determination of pH-Optima and Stability

The pH-optimum of *Xgr*DyP was determined for the oxidation of 2,6-DMP, RBlue5, and Mn^2+^ ions at pH values ranging from 2.0 to 7.0 in either sodium citrate buffer for 2,6-DMP and RBlue5 or in sodium malonate buffer (50 mM) for the oxidation of Mn^2+^ ions. The latter was specifically assayed in the presence of MnCl_2_ (0.5 mM) by monitoring the formation of Mn^3+^–malonate complexes at 270 nm (ε_270_ = 11.3 mM^−1^ cm^−1^, [30]). The conversion of DMP (5 mM) was detected at 469 nm (ε_469_ = 27.5 cm^−1^ mM^−1^, [33]) and that of RBlue5 (0.1 mM) at 598 nm (ε_598_ = 8.0 cm^−1^ mM^−1^; [34]). All reactions were started by the addition of H_2_O_2_ (0.1 mM) and followed spectrophotometrically (Cary 50 UV–Vis spectrophotometer) over appropriate time intervals between 10 and 60 s.

### 2.7. Kinetic Parameters

Apparent Michaelis–Menten (*K*_m_) and catalytic constants (*k*_cat_) of the purified *Xgr*DyP were determined spectrophotometrically for the substrates ABTS, 2,6-DMP, RBlue5, and Mn^2+^ under the above-mentioned conditions. Lineweaver–Burk plots were made from the initial rates obtained at varying substrate concentrations while the concentration of the co-substrate was held constant.

### 2.8. Peptide Sequencing

To identify the enzyme-encoding gene from the corresponding genomic data of *X. grammica*, analysis of internal peptides (‘peptide mapping’) of purified *Xgr*DyP was performed at the Helmholtz Center for Environmental Research–UFZ (Department of Molecular Systems Biology, Leipzig, Germany). Proteins were digested from a Comassie-stained SDS gel by trypsin and the peptide lysates were analyzed by nano-LC-MS/MS (Appendix A: Proteomics (Peptide Mapping)).

### 2.9. Phylogenetic Analysis

After identification of the protein-coding sequences, Blast2GO was used to annotate the proteins and to identify the DyP genes in the *Xylaria grammica* genome [26].

A multiple protein sequence alignment was calculated using Clustal Omega 1.2.2, embedded in Geneious Prime 2021. The phylogenetic tree was calculated using a maximum likelihood approach with PhyML 3.0 [35]. Branch support was estimated by bootstrapping.

### 2.10. Homology Modeling

Homology modeling of protein 3D structures was performed based on the amino acid sequences of the *Xgr*DyP gene 488 using the webserver C-I-TASSER [36]. For modeling, the proposed Mn^2+^ oxidation site, a DyP from *P. ostreatus* (r*Pos*DyP4, PDB 6fsk chain A) was used as a template. The resulting protein models were superimposed and adjusted using PyMOL (The PyMOL Molecular Graphics System, Version 2.2.0 Schrödinger, LLC; http://pymol.org/ accessed on 18 September 2021) with the above-mentioned template sequence.

## 3. Results

### 3.1. Production and Purification of a DyP from X. grammica

Small-scale production of MiP reached the highest levels of up to 750 U L^−1^ during growth in soy-peptone-based complex liquid media. Enzyme secretion was accompanied by alkalization of the medium from pH 5.0 on day 8 to 8.9 on day 20. In parallel, the MiP activity increased from 80 to 750 U L^−1^. (Figure 1). It should be noted that ‘true’ extracellular peroxidase activity (neither of DyP nor of any other heme peroxidase) has not yet been described for an ascomycetous fungus. Therefore, the *X. grammica* peroxidase (*Xgr*DyP) was chosen for further studies regarding enzyme purification and characterization.

Production of *Xgr*DyP on a larger scale was performed in 500 mL flasks containing 200 mL SPM medium. After reaching a sufficient activity level (up to 2028 U L^−1^ on day 26 of cultivation; data not shown), the culture liquid was harvested, filtrated, frozen, and stored at −20 °C until enzyme purification. The freezing and thawing step allowed the removal of precipitating polysaccharides that were separated by a second filtration step. After the concentration of the crude enzyme was obtained, *Xgr*DyP was purified by several steps of FPLC using AEX and SEC techniques.

By using Q-Sepharose as the first separation step, the crude enzyme extract was divided into two fractions: an unbound and a bound fraction. The latter was used for further purification studies. It was re-chromatographed by a second Q-Sepharose step under the same conditions (Q-Sepharose_II) and thus, almost 100% enzyme recovery (1160 U) could be achieved. In the next step, the fraction with the highest peroxidase activity was loaded on an SEC column. Thereby further protein material was separated from the target protein (*Xgr*DyP) without noticeable activity loss, which led to an increase in specific activity from 4.6 to 10.9 U mg^−1^ (purification factor increased from 1.4 to 3.2; Table 1).

By using a MonoQ column, the active SEC fraction was divided into two peaks, which was accompanied by an activity loss of approximately 44% in relation to the preceding step and with a further increase in specific activity (up to 14 U mg^−1^). Immediately following, the fraction with the highest activity was purified to apparent homogeneity by a re-chromatographic step on a MonoQ column under almost identical conditions (except that the pH was slightly reduced from 6.0 to 5.8; Figure 2).

Finally, a 15-fold purification was achieved along with a moderate activity recovery of 6% for the homogeneous *Xgr*DyP protein preparation. The final specific activity was 51.0 U mg^−1^, the Reinheitszahl 1.10 and the residual activity amounted to 120 total units corresponding to approximately 2.4 mg total protein (Table 1).

### 3.2. Characterization of the X. grammica DyP

Purified *Xgr*DyP exhibited the characteristic reddish color of heme-containing enzymes and had absorption maxima at 406, 501, and 630 nm (Figure 3A). The protein appeared as a single protein band with a molecular mass of 49 kDa in the SDS-PAGE gel indicating its pure nature (Figure 3B).

In addition to typical peroxidase substrates such as heterocyclic ABTS and phenolic 2,6-DMP, *Xgr*DyP also oxidized non-phenolic VA (data not shown) and Mn^2+^ ions. To our best knowledge, the oxidation of the latter substrate has not been reported for any wild-type DyP nor for any ascomyceteous peroxidase. Against this background, the pH dependencies of *Xgr*DyP for the oxidation of the classic peroxidase substrate 2,6-DMP, the specific DyP substrate RBlue5, as well as for the ‘untypical’ DyP substrate Mn^2+^ were determined. The oxidation of the phenolic substrate (2,6-DMP) occurred with a distinct pH maximum at 3.5, and with approximately 95% and 90% of the maximum activity at pH 3.0 and 4.0, respectively (Figure 4). Above pH 4.0, however, 2,6-DMP oxidation dropped rapidly to 20% at pH 5.0 and to almost zero at pH 6.0; on the other hand, over 40% activity remained at pH 2.0. The pH optimum of the RBlue5 oxidation formed a sharper maximum at pH 4.0 but otherwise resembled the 2,6-DMP profile.

The curve for the oxidation of Mn^2+^ showed a maximum at pH 4.5 and appreciable activities down to pH 4.0 and up to pH 5.5 (80–90% in relation to the maximal activity). Interestingly, Mn^2+^ oxidizing activities were still detectable at less optimal pH above 5.5.

### 3.3. Kinetic Parameters of the Purified X. grammica DyP

The Michaelis–Menten constants (*K*_M_) for ABTS, 2,6-DMP and RBlue5 were calculated to be 41, 12, and 41 µM, which is in the range of those of the DyP from *A. auricula-judae* (*Aau*DyP, [8]). The enzyme showed the highest affinity (*K*_m_ = 12 µM) and catalytic efficiency (*k*_cat_/K_M_ = 2499 s^−1^ mM^−1^) for the phenolic substrate 2,6-DMP. The *K*_M_ value for the oxidation of Mn^2+^ ions (49 µM) was relatively low (suggesting high affinity to the substrate) compared to recombinant DyPs from *P. ostreatus*, and interestingly ranged in the same order of magnitude as those of true MnPs and VPs of *Pleurotus* spp. and other white-rot fungi [12]. On the other hand, the catalytic efficiency and turnover number of *Xgr*DyP for Mn^2+^ ions (*k*_cat_/*K*_M_ = 8.0 s^−1^ mM^−1^ and *k*_cat_ = 0.4 s^−1^) were notably lower than the respective data for typical Mn-oxidizing peroxidases from basidiomycetous fungi (Table 2).

### 3.4. XgrDyP Encoding Genes and Their Deduced Protein Sequences

A custom blast search using reference sequences from *B. adusta* and *Pseudomonas aeruginosa* DyPs indicated the presence of three DyP-encoding genes (*g488*, *g9177*, and *g195*6) in the *X. grammica* genome (Figure 5).

The properties of the three genes (e.g., ORF and the number of introns) and their deduced putative protein sequences (protein length, molecular weight, etc.) are given in Appendix A. The two genes *g488* and *g9177* had the typical conserved heme-binding motif G-X-X-D-G [5]; by contrast, *g1956* had an unusual G-X-X-D-H motif (Appendix A). By peptide de novo sequencing, it turned out that the purified *Xgr*DyP is a product of *g488* (Figure 5). The deduced protein from DyP *g488* contained 493 amino acids, had a hypothetical molecular mass of 53.9 kDa and a pI of 6.5. No signal peptide was predicted for all three DyP sequences (Appendix A).

The phylogenetic tree (Figure 5) contains sequences from 34 selected fungal DyPs from basidio- and ascomycetes. It demonstrates that the *Xgr*DyPs encoded by three genes belong to different clades. For instance, the gene encoding the purified protein *g488* matches within group ‘D’, in which all so far characterized basidiomyceteous DyPs (e.g., DyPs from *B. adusta*, *Bad*DyP and *A. auricula-judae*, *Aau*DyP) can be grouped [22] or within the new group ‘V’ according to Yoshida and Sugano [21].

The *Xgr*DyP encoded by *g488* showed the highest homologies to hypothetical DyPs from phylogenetically related ascomycetes (e.g., to 79% to *Xylaria hypoxylon* and to 78% to *Xylaria multiplex*). The highest homologies to characterized DyPs were found for the recombinant protein from *P. ostreatus* DyP4 (r*Pos*DyP4; ~44%) and the wild-type proteins from *B. adusta* (*Bad*DyP; ~40%) and *A. auricula judae* (*Aau*DyP1; ~38%).

Gene *g9177* could be assigned to group ‘B’ [22] or the new group ‘P’ according to Yoshida and Sugano [21], which mainly contains bacterial DyPs or encapsulin encoding sequences and a putative DyP from the wood-dwelling ascomycete *X. multiplex* (KAF2971125). With the latter, *g9177* shares 85% homology. The gene *g1956*, however, proved to be an unusual gene encoding probably for both a DyP and a pyruvate-formate lyase-like domain (similar genes are deposited in NCBI under GAW13804.1 and GAP86733.1). Moreover, it groups within a cluster of four putative ascomyceteous DyPs with one, e.g., from *Rosellina necatrix* (with 78% homology; Figure 5).

## 4. Discussion

### 4.1. Production of a New DyP-Type Peroxidase from the Ascomycete X. grammica

Fungal DyPs have been reported from more than 100 basidiomycetous and ascomycetous species with about 400 and 200 individual sequences, respectively. Of them, nine proteins from seven organisms have been characterized so far (*B. adusta*, *M. scorodonius*, *Termitomyces albuminosus* (formerly known as *G. candidum*), *A. auricula-judae*, *E. glandulosa*, *M. epipterygia*, *I. lacteus*; [37]); moreover, there are numerous reports on intracellular bacterial DyPs, which, however, differ considerably from their fungal counterparts [21].

The conditions under which DyPs are expressed and secreted are not yet fully understood [37,38]. The physiological function of this enzyme type also remains unclear. The induction of DyPs under lab-scale conditions is usually accomplished by using complex plant-based growth media (containing a variety of plant ingredients including secondary metabolites). The fungal DyPs that have been characterized so far (only nine representatives) can be produced in soybean meal suspension or in diluted ‘organic tomato juice’ supplemented with ‘elicitors’ such as phenols or terpenoids (as shown for the DyPs or *M. epipterygia* and *E. glandulosa*; [9]). *X. grammica* is the first ascomycetous fungus that has been shown to produce a wild-type DyP. Enzyme levels were up to ~750 U L^−1^, which is sufficient for protein purification and characterization studies. The enzyme titers secreted by the fungus correspond to medium activity levels. In contrast, several other fungi have been found to secrete just low amounts of DyP (<100 U L^−1^), which is sometimes difficult to verify and usually not sufficient to serve as starting point for purification studies (e.g., DyPs of *Mycena haematopus* and *Stropharia rugosoannulata*; [39]). On the other hand, there are a few fungi (species and strains) that secrete larger amounts of enzyme, enabling purification and subsequent characterization and application studies possible, e.g., *A. auricula-judae* with ~8000 U L^−1^ DyP [8].

### 4.2. Comparison of XgrDyP with Other Fungal DyP-Type Peroxidases

The DyP of *X. grammica* was purified using FPLC protocols (AEX, SEC) similar to those that have already been successfully used to prepare other fungal wild-type DyPs (e.g., from *A. auricula-judae*, *M. epipterygia*, and *E. glandulosa*; [8,9]). Finally, one homogeneous *Xgr*DyP fraction was obtained that could be used for enzyme characterization. All purified and characterized reference DyPs are of basidiomycetous origin and as of yet, there are no enzymes of this type available from the ascomycetes. The specific activity (51 U mg^−1^ with ABTS as substrate), molecular mass (49 kDa), and spectroscopic absorbance maxima (with the characteristic Soret band at 406 nm) of this ascomycetous DyP are in the range of data reported for its basidiomycetous counterparts (up to 469 U mg^−1^, 43 to 69 kDa, Soret bands between 405 and 407 nm; [9]).

### 4.3. Catalytic Properties of X. grammica DyP 

The substrate spectrum of *Xgr*DyP is strongly indicative of its affiliation to the protein family of DyP-type peroxidases (including RBlue5 oxidation; [4]). Remarkably, the catalytic constants for certain substrates indicate that *Xgr*DyP shares characteristics of both DyP-type peroxidases and high-redox potential class II PODs such as VP or MnP (e.g., from *Pleurotus eryngii*, *P. ostreatus* or *B. adusta*) or LiP (e.g., from *P. chrysosporium*). Thus, the *K*_M_ values (affinities) for the oxidation of typical peroxidase substrates such as ABTS and 2,6-DMP by *Xgr*DyP are in the same range as the values reported for two *Aau*DyPs [8] and the oxidation of the non-phenolic aromatic substrate veratryl alcohol occur over the same range of pHs (pH < 3.0) as those reported for *Aau*DyPs [9]. Furthermore, the affinities of *Xgr*DyP to these ‘classic’ peroxidase substrates were higher than those described for the MnPs of *P. ostreatus Pos*MnP3 and *Pos*MnP6 (two short hybrid-type manganese peroxidases (hMnPs) that also oxidize, in addition to Mn^2+^, phenolics and ABTS [5]; *K*_M_ = 778 and 1020 µM, respectively for ABTS and *K*_M_ = 59,000 and 117,000 µM, respectively for 2,6-DMP). Moreover, the recombinant DyPs from *P. ostreatus* (r*Pos*DyP1 and 4) had much lower affinities for these typical peroxidase substrates (*K*_M_ = 779 and 787 µM, respectively for ABTS and *K*_M_ = 31,100 and 126 µM, respectively, for 2,6-DMP; Appendix A) [12].

Interestingly, the ascomycetous DyP belongs—from the biochemical point of view—to the small group of Mn^2+^-oxidizing DyPs, of which five representatives are so far known. Among them are a few bacterial representatives (from *Pseudomonas fluorescens*, *R. jostii*, and *Amycolatopsis* sp.; [11,40,41] and only two recombinant basidiomycetous proteins from *P. osteratus* (r*Pos*DyP1 and 4) expressed in *E. coli* [12]). The affinity of *Xgr*DyP for Mn^2+^ ions (*K*_M_ = 49 µM) is in the typical range of ‘classic’ MnPs (e.g., from *P. chrysosporium*, 4–9 µM, [42]; *Agrocybe praecox* and *Stropharia coronilla*, 17 and 12 µM, respectively, [43]; *P. ostreatus*, 10 µM, [44]; as well as for VPs (e.g., from *P. eryngii*, 12–20 µM; [45])). On the other hand, the turnover numbers (*k*_cat_) of ‘true’ MnPs are (e.g., 150 s^−1^ for *B. adusta*, [46] and or 218 s^−1^ for *P. chrysosporium*, [47]) are two orders of magnitude higher than the *k*_cat_ values determined for the ascomycetous DyP (*k*_cat_ = 0.4 s^−1^). The Michaelis–Menten constants of the Mn^2+^-oxidizing fungal DyPs (r*Pos*DyP1 and 4) are much higher (*K*_M_ = 2780 and 286 µM, respectively) than that of *Xgr*DyP, but the catalytic efficiency, at least of r*Pos*DyP4, is about one magnitude higher (196 s^−1^ mM^−1^) than that calculated for the *Xgr*DyP (8 s^−1^ mM^−1^). All these findings indicate that Mn^2+^ ions bind with high affinity to the *Xgr*DyP protein (very probably at suitable acidic amino acid, i.e., aspartates and glutamates) but their subsequent oxidation into reactive Mn^3+^ may proceed with much lower efficiency compared to MnPs and VPs. Mn^2+^ oxidation by *Xgr*DyP proceeded most effectively under acidic pH conditions, which is also a characteristic feature of all ‘classic’ MnPs and comparable to the respective activities of certain recombinant DyPs from *P. ostreatus* (e.g., at pH 4.5; [12]).

Generally, the ability to efficiently oxidize Mn^2+^ ions is a specific catalytic property of certain class II PODs (MnP, hMnP (hybrid-type MnP), and VP). These enzyme types are exclusively found in Basidiomycota (e.g., in the families Polyporaceae, Corticiaceae, Pleurotaceae, Agaricaceae, or Strophariaceae) causing white-rot or accomplishing soil-litter decomposition [5,48]. The high abundance of these peroxidases among saprotrophic basidiomycetes and their ubiquitous presence in natural deadwood strongly suggests that the oxidation of manganese (Mn^2+^→Mn^3+^) is one of the key steps in lignin decomposition [5,49,50,51].

In contrast, according to comprehensive genomic data, wood-inhabiting Ascomycota (e.g., Xylariaceae) are obviously lacking MnPs and other high-redox potential class II PODs [52,53,54]. In this context, the finding of an Mn-oxidizing ascomycetous DyP is remarkable, and could explain why some of these fungi (e.g., *X. grammica* or *Xylaria polymorpha*) can nevertheless degrade and mineralize lignin (e.g., β-*O*-4 dimers such as adlerol), at least to some extent [8,23,55]. In other words, certain DyPs may fulfill the roles of MnPs/VP in some ascomycetous fungi, for example, in those species causing a strong soft rot [12].

### 4.4. Structural Aspects

As other peroxidases of this type, *Xgr*DyP *g488* has a characteristic heme environment comparable with that of some basidiomycetous DyPs (e.g., sequence identity to *Aau*DyP1 38%, [6,18] and to r*Pos*DyP4 44%, [56]). It contains a proximal histidine (His336), which is structurally homologous to His361 and His334 in r*Pos*DyP1 and 4, respectively [12], as well as to His306 and His304 in *Bad*DyP and *Aau*DyP, respectively [18]. The distal heme environment of *Xgr*DyP *g488* contains conserved amino acids that act as a proton acceptor (aspartate, Asp192) and charge stabilizer (arginine, Arg359; [5,6,18]) as well as the co-substrate binding/guiding residues (phenylalanine/Phe390 and leucine/Leu388; [37]). The entrance to the heme channel of *Xgr*DyP *g488* is characterized by the presence of an aspartate, an amino acid residue previously described for *Aau*DyP1 as a “gatekeeper.” Flipping of the Asp168 side chain of *Aau*DyP1 results in molecular movement and thus some widening of the heme entrance channel [18].

Identification of a hypothetical Mn^2+^ oxidation site in the hypothetical 3D structure of *Xgr*DyP *g488* was performed by high-resolution protein modeling using the C-I-Tasser webserver. The C-I-Tasser calculation identified r*Pos*DyP4 (PDB: 6fsk) as the protein structure most closely resembling *Xgr*DyP, with a TM-score of 0.92 (data not shown). Therefore, the tertiary structure of recombinant Mn^2+^-oxidizing r*Pos*DyP4 was used to construct the hypothetical 3D model of *Xgr*DyP *g488*.

In contrast to the Mn^2+^-binding amino acids in classical MnPs (e.g., from *P. chrysosporium*, *Pch*MnP, [57]) which comprise carboxylic groups of mostly three acidic amino acids (i.e., glutamates or aspartates) and a heme propionate residue, no similar binding site could be detected in the hypothetical *Xgr*DyP *g488* 3D protein structure (data not shown).

Fueyo et al. [56] identified an Mn^2+^ binding site on the surface of r*Pos*DyP4 based on mutation studies. This consists of the four acidic amino acids (Asp215, Asp352, Asp345, Glu354) and Tyr339. The enzyme obviously uses an electron transfer mechanism (‘electron hopping’) from the glutamate (Glu345) to the heme via an aromatic residue (Tyr339, Figure 6A; [56]). In the case of *Xgr*DyP, no similar surface-exposed Mn^2+^ oxidation site could be identified within the structure model. In the corresponding region, only two rather distantly related acidic amino acids are present (Glu353 and Asp347). Their probably coordinating carboxylates do not show a distinct orientation towards the protein surface and no aromatic amino acid is available for an electron transfer towards the heme (Figure 6B).

In addition, another potential Mn^2+^ binding site on the protein surface was described in the same r*Pos*DyP4 and in the crystal structure of the Mn^2+^-oxidizing DyP2 of the bacterium *Amycolatopsis* sp. (*Asp*DyP2; PDB: 4g2c). Again, these are nearby acidic and aromatic amino acids (Figure 7, e.g., Asp260, Asp275, Asp285, and Tyr190 in *Asp*DyP2; [40,56]).

In the model structure of *Xgr*DyP *g488*, a comparable region near the protein surface was detected. The Mn^2+^ ion could be coordinated by three acidic amino acids (Asp269 and 288, and Glu265; Figure 7A) and the oxidation may be supported by four aromatic amino acids (three Phe190, 264, and 188 and one Trp298), which act as partners for the electron transfer. The three Phe residues are in close proximity to each other (e.g., 2.4 Å between Phe264 and 190) as well as to the heme center, making them suitable for electron transport. This protein environment is rather similar to the Mn^2+^ binding site postulated for the bacterial *Asp*DyP2. It is also interesting to note that the relevant acidic and aromatic amino acids are conserved in both bacterial and several fungal DyPs [56].

Finally, it should be mentioned again that the substrate oxidation site of *Xgr*DyP *g488* is putative and that the orientation and distances between the involved ligands and metal ions may alter during binding and/or catalysis (Figure 7A). Definite clarification of these points can be only achieved once the crystal structure of *Xgr*DyP (encoded by *g488*) is solved, which will allow an analysis of the molecular architecture of the Mn^2+^ coordination sphere.

## 5. Conclusions

In the future, recombinant expression studies, although challenging, will facilitate and improve the production of target heme peroxidases including DyPs. This, in turn, would be an important prerequisite for comprehensive protein structure–function and molecular engineering studies. Both approaches may help to identify catalytically relevant amino acids (e.g., for the binding and oxidation of Mn^2+^ by *Xgr*DyP). Whether the Mn^2+^ oxidizing capability of *Xgr*DyP really has physiological implications for wood and lignin degradation by *Xylaria* spp., remains unclear. Possibly, ‘DyP-knock-out mutants’ of *X. grammica* (obtainable using modern techniques of gene-editing such as CRISPR/Cas) will help to answer this important question.

Not least, testing the lignocellulose degrading activities of the *Xgr*DyP and *X. grammica* will help to answer the ecologically and physiologically relevant question as to whether manganese oxidation is actually the basis of ligninolysis. In particular, evidence for a DyP/Mn^2+/3+^-based depolymerization system would further improve our understanding of lignin degradation in nature and shed new light on the role of Ascomycota in this process.

## Figures and Tables

**Figure 1 biomolecules-11-01391-f001:**
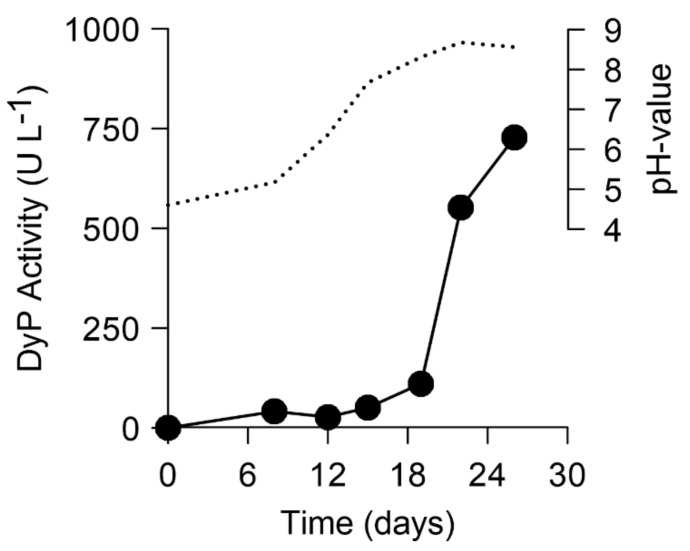
Time course of DyP production (black circles) by *X. grammica* in SPM medium; activity was measured with ABTS and H_2_O_2_, dotted lines—pH value.

**Figure 2 biomolecules-11-01391-f002:**
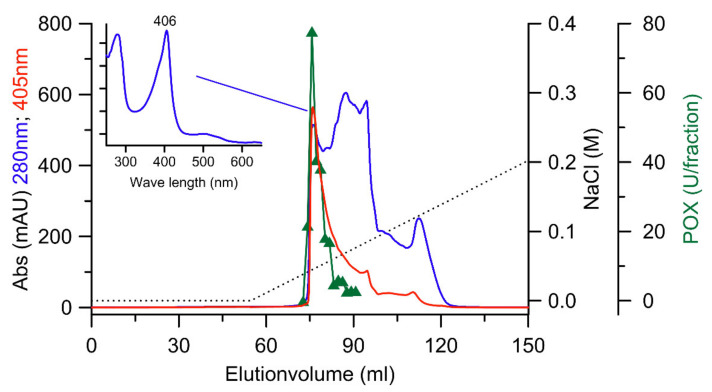
FPLC elution profiles of the last re-chromatographic purification step of *Xgr*DyP using a MonoQ column (10 × 100 mm). Absorption (mAU) at 280 nm (blue→total protein) and 405 nm (red→heme), DyP activity (green), NaCl gradient (dashed line).

**Figure 3 biomolecules-11-01391-f003:**
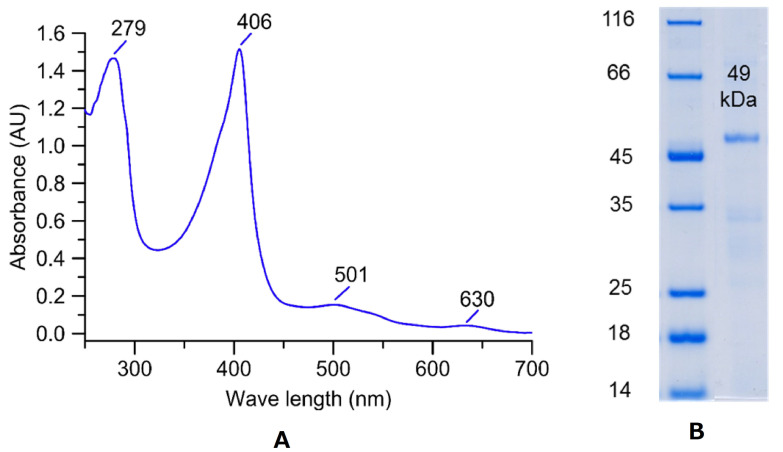
(**A**) UV–Vis absorption spectrum of purified *Xgr*DyP in its resting state with the Soret band at 406 nm and additional bands at 279, 501, and 630 nm representing the δ-, β-, and α-bands, respectively. (**B**) SDS-PAGE of the purified *X. grammica* DyP (*Xgr*DyP; right), left: protein marker.

**Figure 4 biomolecules-11-01391-f004:**
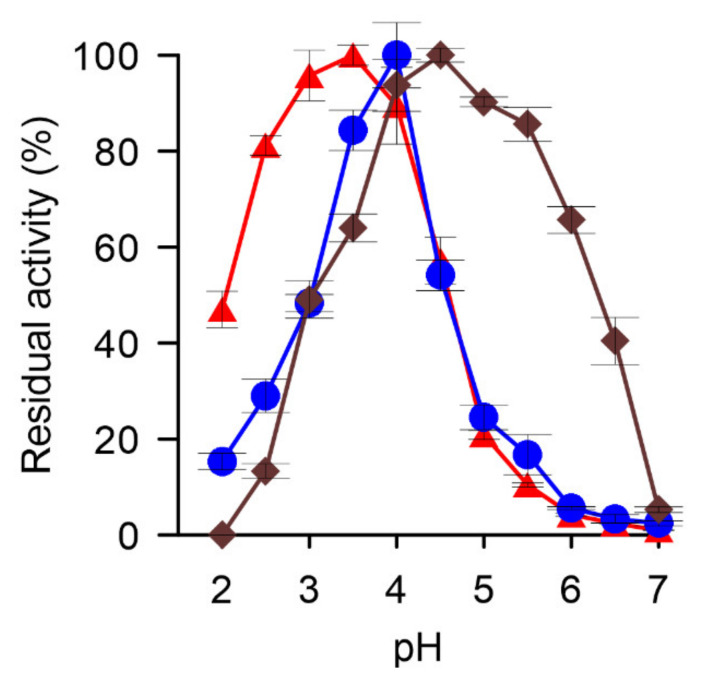
pH-Optima for the oxidation of 2,6-DMP (red triangles), RBlue5 (blue circles), and Mn^2+^ ions (brown squares) by the purified *Xgr*DyP.

**Figure 5 biomolecules-11-01391-f005:**
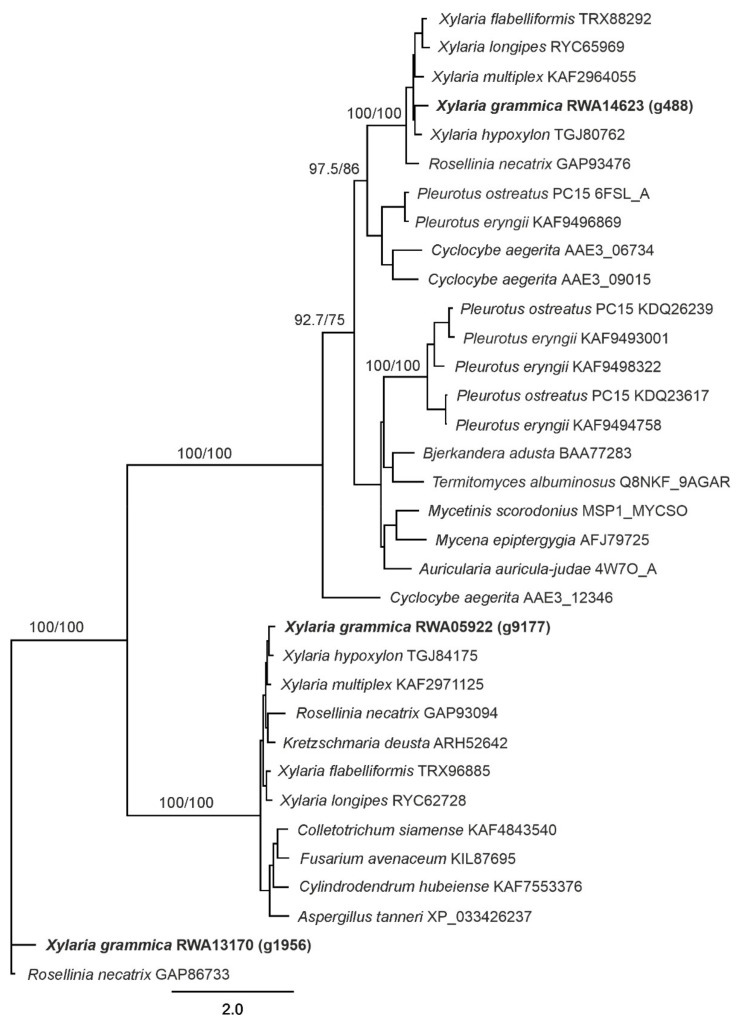
Maximum likelihood phylogenetic tree (log-likelihood: −18,869.58452) using a DyP protein sequence alignment in PhyML 3.0 [35] with the following model: LG+I+G. Branch support was estimated using a bootstrap approach, first value: 2000 replicates of neighbor-joining trees using Jukes–Cantor distances, second value: 100 replicates of the maximum likelihood approach using LG + I + G model.

**Figure 6 biomolecules-11-01391-f006:**
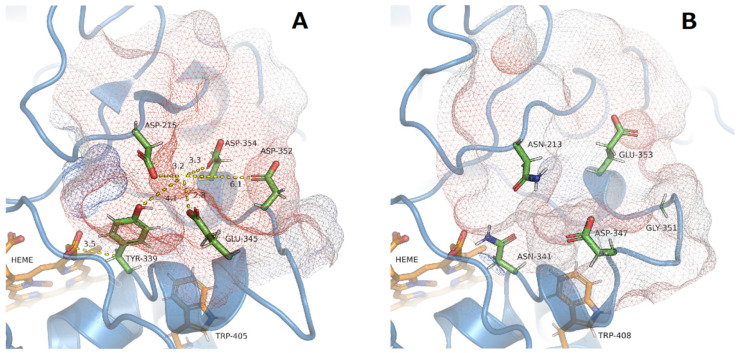
(**A**) Characterized Mn^2+^ binding site and distribution of the coordinating acid amino acids and the electron transferring Tyr339 in r*Pos*DyP4 [56] (distances in Å are indicated yellow dotted lines) and (**B**) the corresponding analogous region in the hypothetical 3D structure of *Xgr*DyP *g488*. The solvent excluded surface (SES) around the putative Mn^2+^ binding site is given as a colored mesh from blue to red according to the electrostatic charge distribution (blue-positive, red-negative) and is used to approximate the hypothetic Mn^2+^ position.

**Figure 7 biomolecules-11-01391-f007:**
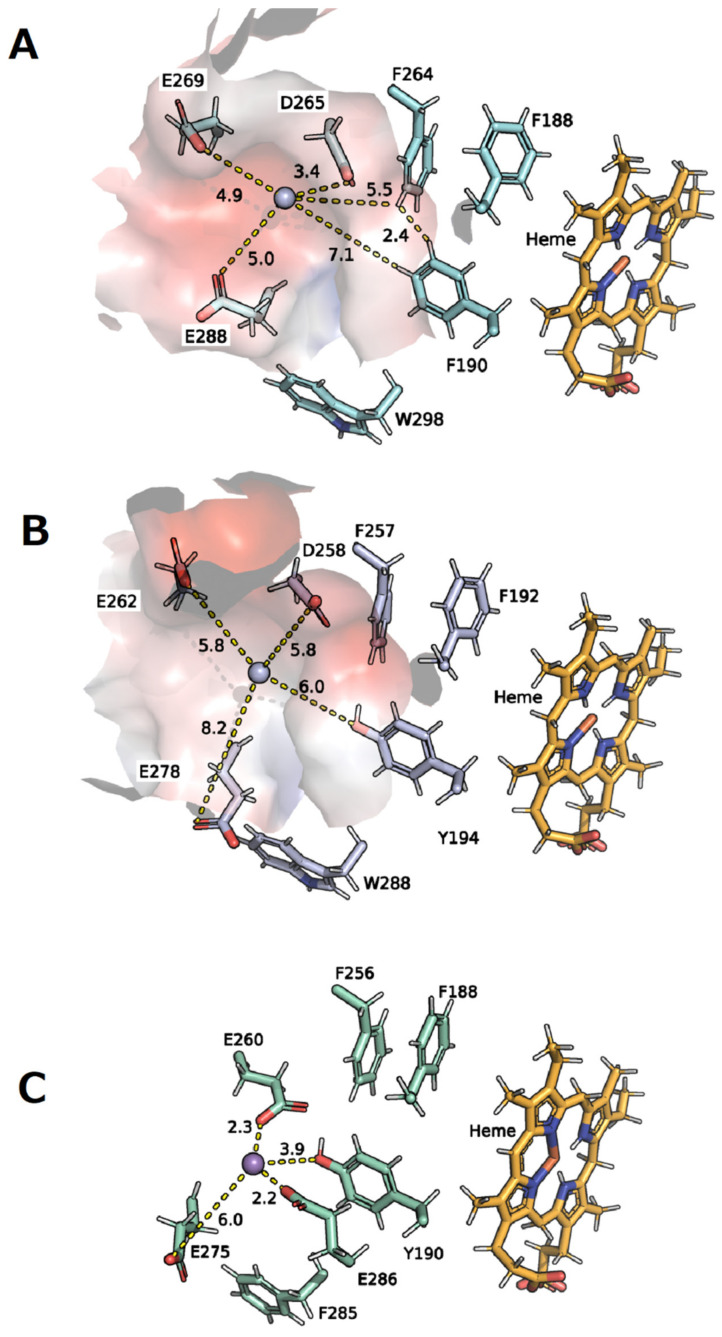
(**A**) Hypothetical Mn^2+^ binding site in the protein model of *Xgr*DyP *g488*, in comparison to the characterized structures of the (**B**) fungal r*Pos*DyP4 (PBD: 6fsk; [56]) and (**C**) the bacterial *Asp*DyP2 (PBD: 4g2c).

**Table 1 biomolecules-11-01391-t001:** Purification of the extracellular peroxidase from *X. grammica* (*Xgr*DyP).

Purifications Step	Activity (U)	Specific Activity (U mg^−1^)	Protein Amount (mg)	Yield (%)	Purification (x-Fold)
Culture filtrate	2028	3.4	595.7	100	-
Ultrafiltrate 10 kDa	1923	3.3	584.0	95	1.0
Q-Sepharose_I	1175	4.4	265.2	58	1.3
Q-Sepharose_II	1160	4.6	254.4	57	1.4
SEC	1301	10.9	119.9	64	3.2
MonoQ_I	568	13.8	41.1	28	4.1
MonoQ_II	120	51.0	2.4	6	15.0

**Table 2 biomolecules-11-01391-t002:** Apparent kinetic constants (*K*_M_, *k*_cat_, and *k*_cat_/*K*_M_) of purified *Xgr*DyP for the substrates ABTS, 2,6-DMP, RBlue5, and Mn^2+^.

Substrate	*k_cat_* (s^−1^)	*K*_M_ (µM)	*k*_cat_/*K*_M_ (s^−1^ mM^−1^)
ABTS	12	41	287
2,6-DMP	29	12	2499
RBlue5	20	41	495
Mn^2+^	0.4	49	8

## Data Availability

Full data sets are available either in public data bases e.g., NCBI, PDB, in the Appendix A (see the main text) or upon request from the corresponding author.

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
