# Peer review of "First Dye-Decolorizing Peroxidase from an Ascomycetous Fungus Secreted by *Xylaria grammica"

_biomolecules, 2021, doi:10.3390/biom11091391_

Round 1

Reviewer 1 Report

See attached

Reviewer 2 Report

The authors of the present work report the discovery and biochemical characterization of a dye-decolorizing peroxidase (DyP) from Xylaria grammica, which is the first DyP from an ascomycete fungus to be isolated and biochemically characterized. The experiments are adequately designed and performed and the manuscript is in overall well written. Please see below my comments/corrections.

  1. Line 230: according to figure 1, DyP production has not reached a maximum on day 24, so this sentence should be corrected accordingly.
  2. Line 234: what do the authors mean by “true extracellular peroxidase activity”? this should be explained in the manuscript
  3. Lines 257-278: The authors should provide a reference or experimental evidence that the precipitate contained polysaccharides.
  4. Line 261: replace “medium” by “step”
  5. Lines 262-263: I assume the authors mean that the “unbound” fraction was re-chromatographed. Please correct accordingly. In addition, in Table 1, the total enzyme activity (from both Q colums) should be shown (correct the number 1,160). It is impossible that the total enzyme activity increases after the SEC step (1,301 Units).
  6. Line 306: SDS-PAGE electrophoresis cannot show the oligomeric state of a protein as it is held under denaturating conditions. The oligomeric state of the protein can be shown through SEC.
  7. Line 412: The authors could comment on the discrepancy between the theoretical (53.9 kDa) and estimated MW (49kDa).
  8. Figure 5: 3 of the 4 homologs of g1956, mentioned in the manuscript (line 431), are not shown in the bottom clade.
  9. Paragraph 4.4: the authors should present the multiple sequence alignment that they used to identify the XgrDyP aminoacids involved in the heme environment, highlighting the “conserved” aminoacids.
  10. Line 537: please indicate the sequence identity between the 2 proteins.
  11. Line 539: I assume the authors mean “construct” instead of “evaluate”.
  12. Line 544-5: the authors of this work used additional techniques to identify the Mn binding site, not only mutations.
  13. Figure 6: in (A), the authors should specify what the yellow dotted lines show. The should display the Mn ion. They should also state what is the mesh that they display in both A and B.
  14. Line 551: the orientation of the side chains can change upon ion binding so this statement is irrelevant and should be removed.
  15. All abbreviations and microorganism names should be written “in full” the first time they appear in the manuscript, eg line 34: DyP, line 70: MnPs and VP, line 50: P. ostreatus etc
